# Neonatal Transport in the Practice of the Crews of the Polish Medical Air Rescue: A Retrospective Analysis

**DOI:** 10.3390/ijerph17030705

**Published:** 2020-01-22

**Authors:** Ewa Rzońca, Stanisław Paweł Świeżewski, Robert Gałązkowski, Agnieszka Bień, Arkadiusz Kosowski, Piotr Leszczyński, Patryk Rzońca

**Affiliations:** 1Chair and Department of Development in Midwifery, Faculty of Health Sciences, Medical University of Lublin, 4–6 Staszica St., 20-081 Lublin, Poland; eva.rzonca@gmail.com (E.R.); agnesmbien@gmail.com (A.B.); 2Department of Emergency Medical Services, Faculty of Health Science, Medical University of Warsaw, 81 Żwirki i Wigury St., 02-091 Warsaw, Poland; stanislaw.swiezewski@gmail.com (S.P.Ś.); r.galazkowski@lpr.com.pl (R.G.); 3Department for Uniformed Services, Main Office of the National Health Fund in Warsaw, 186 Grójecka St., 02-390 Warsaw, Poland; ak.gabi@o2.pl; 4Faculty of Medical Sciences and Health Sciences, Siedlce University of Natural Sciences and Humanities, 2 Konarskiego St., 08-110 Siedlce, Poland; piotr.leszczynski@uph.edu.pl; 5Department of Emergency Medicine, Faculty of Health Sciences, Medical University of Lublin, 4–6 Staszica St., 20-081 Lublin, Poland

**Keywords:** Helicopter Emergency Medical Service, Emergency Medical Service, neonates, newborn infant, interhospital transport, incubator

## Abstract

The aim of the study was to present characteristics of patients transported in incubators by crews of Helicopter Emergency Medical Service (HEMS) and Emergency Medical Service (EMS) of the Polish Medical Air Rescue as well as the character of their missions. The study was based on the method of retrospective analysis of neonatal transports with the use of transport incubators by the crews of HEMS and EMS of the Polish Medical Air Rescue. The study covered 436 medical and rescue transports of premature babies and full-term newborns in the period between January 2012 and December 2018. The study group consisted mainly of male patients (55.05%) who, on the basis of the date of delivery, were qualified as full-term newborns (54.59%). During the transport their average age was 37.53 (standard deviation, SD 43.53) days, and their average body weight was 3121.18 (SD 802.64) grams. A vast majority of neonatal transports were provided with the use of a plane (84.63%), and these were medical transports (79.36%). The average transport time was 49.92 (SD 27.70) minutes with the average distance of 304.27 km (SD 93.05). Significant differences between premature babies and full-term newborns were noticed in terms of age and body weight at the moment of transport, diagnosis based on the International Statistical Classification of Diseases and Related Health Problems (ICD-10), the most commonly used medications (prostaglandin E1, glucose, furosemide, vitamins), National Advisory Committee for Aeronautics (NACA) scale rate as well as the mission type and the presence of an accompanying person.

## 1. Introduction

The issue of decreasing the death rate of mothers and children under the age of five has always been one of the major goals of all human health activities both internationally and locally. One may notice a particular significance of limiting the death rate of children under the age of 5, including the death rate of newborns. It has become an important point of interest in recent years and is now a great challenge when it comes to ensuring health and development of human capital in the whole world. According to the international data, the mortality of children up to 28 days of age decreased at a much slower rate than that of children aged from 29 days to 5 years, and the death of newborns is responsible for more than two-fifths of deaths of children under 5 years of age annually [1,2,3,4,5]. In Poland, the mortality rate of newborns and babies has also decreased [6,7]. The main reasons for deaths of newborns and babies are prematurity and inborn defects. It should be stressed that the number of deaths caused by prematurity is decreasing; however, the scope of prematurity has not changed. This means that the group of children requiring both long-term medical care and rehabilitation is increasing [6].

The progress in reducing neonatal death rate is made in perinatal care, which is a component of comprehensive measures aimed at providing both a mother, a fetus and a newborn with appropriate multifaceted care during pregnancy, delivery and in the postnatal period. This care is ensured by the cooperation of various specialists who create interdisciplinary teams. In order to improve the perinatal care in Poland, a three-stage model of perinatal care was created in the 1990s [8,9,10]. In the three-stage model of perinatal care, level I covers health care for a woman during physiological pregnancy, delivery and in the postnatal period as well as for a healthy newborn baby. It also includes providing short-term care in the case of unexpected pathology. Level II covers care in the case of medium-level pathology, while level III aims at providing treatment and care for pregnant women at high risk and in the case of sudden life-threatening conditions [10]. However, due to various reasons, there are situations in which women give birth in places that do not provide proper neonatal care for a newborn child. Then, it is necessary to transport a newborn to a center with an appropriate level in the referral system [10,11].

The transport of a newborn whose life is threatened to a center with a higher level in the referral system in Poland is performed by a ground-based specialist N ambulance or by Helicopter Emergency Medical Service (HEMS) and Emergency Medical Service (EMS) of medical air rescue [10,12]. Ground teams of specialist N ambulances consist of three people (a doctor, a nurse/midwife, a driver), and the ambulance is equipped with the necessary equipment. It also meets the relevant technical and quality standards compliant with national and European standards [10]. However, if a newborn is transported over a distance exceeding 100 km, the transportation should be provided by air transport teams [10,13]. In Poland, neonatal transports of premature infants and full-term newborns weighing not more than 5000 g are carried out by Medical Air Rescue crews [10,12]. Newborn transport is carried out in a direct mode, namely bed-to-bed (B2B) mode, in which the patient is picked up directly from the ordering hospital and handed over in the target hospital in the same way, and in combined mode (nB2B), in which HEMS crews cooperate with ground-based teams. The Helicopter Emergency Medical Service and Emergency Medical Service crews are equipped with the necessary medical equipment and means to provide specialist care during transport to a hospital. They have four transport incubators available. The HEMS crew consists of three people: A professional pilot, a doctor and a paramedic or a nurse. The EMS crew consists of four people: Two professional pilots, a doctor and a paramedic or a nurse. Neonatal transports are provided by two types of aircrafts: Piaggio P.180 Avanti and EC 135 helicopter. Today, Polish Medical Air Rescue consists of 21 regional HEMS bases, one seasonal HEMS base (in Koszalin) and one EMS transport team located at the Warsaw Frederic Chopin Airport (Figure 1) [12,14].

It was precisely the issue of the distance and the necessity of transport that contributed to the authors’ study aimed at presenting the characteristics of both the patients transported in incubators of the Polish Medical Air Rescue crews and the missions themselves.

## 2. Materials and Methods

### 2.1. Study Design

The study was based on the method of retrospective analysis of neonatal transports with the use of an incubator by the crews of HEMS and EMS of Polish Medical Air Rescue. The study covered all medical and rescue transports of premature babies and full-term newborns with a birth weight not exceeding 5000 g in the period between January 2012 and December 2018 in the whole territory of Poland. The exclusion criteria were cases in which transport orders were cancelled, the mission was not undertaken due to bad weather conditions or technical problems and situations in which the doctor of the transport crew disqualified the patient due to too severe clinical condition. After exclusion of cases in accordance with the exclusion criteria, 436 cases of neonatal transports were qualified for the final analysis. The chief executive officer of the Polish Medical Air Rescue granted his consent to conduct the study, while the retrospective nature of the study did not require the consent of the Bioethics Committee.

The study was carried out on the basis of medical and operational documentation of HEMS and EMS crews of Polish Medical Air Rescue. During the process of the documentation analysis the following information was obtained: The date and destination, the data concerning characteristics of transported patients, main diagnoses based on the International Statistical Classification of Diseases and Related Health Problems (ICD-10), parameters of clinical condition of patients, medical emergency procedures, medications used during the transport as well as the information related to the characteristics of the completed transportations.

The National Advisory Committee for Aeronautics (NACA) scale was used to compare the clinical condition of patients. This is a commonly used scale to assess the severity of injury or disease in pre-hospital conditions in Western European countries. This scale distinguishes 8 groups of patients according to the impairment of life-supporting functions and symptoms of injury, illness or poisoning:NACA 0—no body injuries or illnesses;NACA 1—body injuries or illnesses in case of which medical care is not necessary;NACA 2—body injuries or illnesses that require examination and treatment, but with no need of hospitalization;NACA 3—body injuries or illnesses with no acute life threat but requiring hospitalization;NACA 4—body injuries or illnesses that may lead to deterioration of vital signs;NACA 5—body injuries or illnesses posing a great threat to life;NACA 6—body injuries or illnesses leading to sudden cardiopulmonary arrest;NACA 7—fatal body injuries or illnesses [15,16].

### 2.2. Statistical Analysis

The data obtained in the process of documentation analysis were analyzed statistically with the use of STATISTICA program, version 13.2 (StatSoft, Cracow, Poland). The quantity (*n*) and percentage (%) are used to describe the qualitative data, while mean (M) and standard deviation (SD) are used to describe the quantitative data.

The Shapiro–Wilk normality test was used to verify the normality of the distribution of quantitative variables. The Chi-squared test was adopted in order to assess statistically significant differences between qualitative variables, while the Mann–Whitney non-parametric *U* test was used to analyze the differences between the two independent groups. The odds ratio (OR) was implemented to assess the chance of particular congenital malformations in the examined group of patients. The accepted materiality level was *p* < 0.05.

## 3. Results

The study covered 436 cases of neonatal transports carried out by HEMS and EMS Polish Medical Air Rescue crews. The study group consisted mainly of male patients (55.05%) who, on the basis of the date of delivery, were qualified as full-term newborns (54.59%). The average age of patients during transport was 37.53 days (SD 43.53), and the average body weight of babies during transport was 3121.18 g (SD 802.64). The main diagnoses were congenital heart malformations, which constituted more than two-fifths of all cases (41.28%). The analysis of vital signs and clinical condition of patients during transport showed that the vast majority of patients in incubators were conscious during transport (73.17%), their skin coloring was normal (80.96%), there was no cyanosis (89.91%) and no respiratory effort (78.21%). The average breath count was 36.36 per minute (SD 16.95), saturation 93.55% (SD 6.49), heart rate 141.86 beats per minute (SD 17.27), systolic pressure 141.86 mmHg (SD 17.27), diastolic pressure 45.23 mmHg (SD 10.62) and NACA was 3.66, indicating a moderate to severe clinical condition. The most common medical emergency procedure needed by transported patients was intubation (27.29%), and the most commonly used medications were glucose (44.04%) and prostaglandin E1 (33.03%) (Table 1).

The conducted statistical analysis showed a statistically significant correlation between the classification of the patients based on the date of delivery and age during transport (*p* < 0.0001); body weight during transport (*p* < 0.0001); diagnosis based on the ICD classification 10 (*p* < 0.05); the most commonly used medications: Prostaglandin E1 (*p* < 0.0001), glucose (*p* < 0.05), furosemide (*p* < 0.05) and vitamins (*p* < 0.05) as well as the rate of the NACA scale (*p* < 0.05). The analysis shows that in the group of preterm infants who were older at the time of transport, cardiovascular malformations, congenital malformations of the gastrointestinal tract and respiratory disorders originating in the perinatal period as well as congenital malformations of the nervous system were more common. Glucose and vitamins were administered more often than in the group of full-term newborns. The group of full-term newborns was characterized by higher body weight during transport and more severe clinical condition, as indicated by the NACA scale (*p* < 0.05). These patients received prostaglandin E1 (*p* < 0.0001) and furosemide (*p* < 0.05) more often than premature babies. In other cases no statistically significant dependencies were found (*p* > 0.05) (Table 1).

Table 2 presents the characteristics of missions performed with the use of a transport incubator. The vast majority of neonatal transports were carried out by aircraft (84.63%). These were medical transports (79.36%), mainly in spring (28.90%) and on weekdays (87.39%). In the study group, the patient was most often transported to a tertiary care center (78.21%), and the transport took place without an accompanying person (57.34%). In the analyzed material, the average transport time was 49.92 min (SD 27.70), with an average transport distance of 304.27 km (SD 93.05), and the time from taking care of the patient to patient transfer to the target hospital was 86.29 min (SD 33.57), while the time from the receipt of the call to patient transfer was 269.16 min (SD 118.55).

In the group of the full-term newborns, the transport was more often carried out in the rescue mode (25.63%), and during the transport the patients were accompanied by a parent (47.90%). The stated dependencies were statistically significant (*p* < 0.05). However, there were no statistically significant correlations between the classification of the patients on the basis of the date of delivery and the reference level of the target hospital, time of transport, care and time of the receipt of the call to patient transfer and the transport distance (*p* > 0.05) (Table 3).

Table 4 presents the analysis of the quotient of chances for individual congenital malformations in relation to the classification of the patient on the basis of a delivery date. In the group of preterm infants, the chance quotient was higher for congenital malformations of the cardiovascular system (OR 1.30; 95% CI 0.81–2.08), congenital malformations of the gastrointestinal tract (OR 1.70; 95% CI 0.98–2.93), respiratory disorders originating in the perinatal period (OR 1.29; 95% CI 0.68–2.46) and congenital malformations of the nervous system (OR 2.50; 95% CI 0.92–6.77). In the case of the female sex, however, there is a greater chance of the occurrence of congenital malformations of the gastrointestinal tract (OR 1.17; 95% CI 0.68–2.02), congenital malformations affecting multiple systems (OR 1.07; 95% CI 0.38–3.02) and other malformations (OR 1.32; 95% CI 0.67–2.60).

## 4. Discussion

Since it is not always possible to predict the occurrence of defects, illnesses or premature birth, there is often a need to transfer newborn babies to an appropriate center as late as after their birth [10,11,17]. The transport of a newborn patient in an incubator from a center of lower level in the referral system is carried out to ensure proper diagnostic and therapeutic procedures as well as the care itself in the center of appropriate level in the referral system [10,11,17,18,19,20]. The basic method of neonatal transport in Poland is transportation in ground-based specialist ambulances or by the medical air rescue, especially if the transport distance is to exceed 100 km [10]. Therefore, the distance issue in the case of transporting patients in incubators by the Polish Medical Air Rescue crews contributed to the authors’ study in this area.

The first stage of the study was to analyze the characteristics of the patients transported in incubators. The results of the authors’ own study showed that the patients transported in incubators were mainly males. During the transport, their average age was 37.53 days and their body weight was 3121.18 g, while on the basis of the date of delivery they were qualified as full-term newborns. Moss et al. (2005) in the study on the safety of neonatal transport found out that among the transported patients there were mainly newborns whose average gestational age was 32 weeks and birth weight was 1850 g [20]. The results of the study by Henry and Trotman (2017) on neonatal transport in Jamaica showed that male patients predominated, the average gestational age was 34.9 weeks and birth weight was 2.3 kg [21]. The research carried out by Frid et al. (2018) conducted in Sweden on the basis of the analysis of 187 emergency air transports (acute airborne transports) showed that the average gestational age of the transported babies was 35 weeks (range: 22–42 weeks), and their birth weight was 2627 g (467–5300 g) [22].

The most common reason for transporting patients in incubators by Medical Air Rescue crews in the analyzed material were congenital heart malformations. On the other hand, the study carried out by Frid et al. (2018) showed that the main indications for neonatal transport were therapeutic hypothermia after perinatal asphyxia, extremely preterm birth and respiratory failure [22]. Additionally, in Henry and Trotman’s study (2017), more than half of all neonatal transports were due to respiratory disorders [21]. Kemple et al. (2004) in their study on neonatal transport in England showed that the most urgent reasons for neonatal transfer were the need for surgery, intensive neonatal surveillance as well as cardiac or neurological problems [23]. Yan Leung et al.’s study (2019), which analyzed the transport of infants from regional or private hospitals to the intensive care units of Queen Mary Hospital in Hong Kong, showed that the dominant causes of transport were cardiac problems, the need for surgery and respiratory problems [24].

The issue of premature births, i.e., before the 37th week of pregnancy and prematurity, together with its consequences, is a challenge for governments and health care worldwide [25,26,27,28]. The authors’ own study showed that nearly half of the transported patients in incubators were born prematurely. In the case of preterm infants, the more common reasons for their transport to centers with a higher level in the referral system were congenital malformations of the cardiovascular and gastrointestinal systems, respiratory disorders originating in the perinatal period and congenital malformations of the nervous system. Interesting studies on congenital malformations of newborns, analyzed on the basis of gender and prematurity in the USA, were conducted by Egbe et al. (2015). They showed that male infants had a higher risk of isolated congenital malformations, while preterm infants had a higher risk of isolated, multiple and total congenital malformations. They also found that developmental malformations of heart, as well as the genitourinary, neurological and respiratory system were more frequent in prematurely born neonates; however, the frequency of craniofacial developmental defects was lower [29]. However, Dursun et al. (2014) conducted a study on the occurrence of congenital malformations in one neonatal intensive care unit in Turkey and showed that male newborns dominated among the examined patients and that the most common defect was found in the cardiovascular system [30].

The clinical condition of patients transported in incubators by the Polish Medical Air Rescue crews in the analyzed material was stable. Respiratory rate was 36.36 per minute, saturation 93.55%, heart rate 141.86 beats per minute, systolic pressure 76.60 mmHg, diastolic pressure 45.23 mmHg, while the NACA 3.66 scale value indicated moderate to severe clinical condition of patients. In the study of Bastug et al. (2016) on children transported and treated in the Neonatal Intensive Care Unit of the Erciyes University, Faculty of Medicine in Turkey, the average heart rate after transport was 134 beats per minute in the group of patients weighing 501–1500 g, 145 beats per minute in the group of patients weighing 1501–2500 g and 139 beats per minute in the group of patients weighing above 2500 g. The respiratory rate values were as follows: The group with the body weight of 501–1500 g: 55 breaths per minute, the group with the body weight of 1501–2500 g: 52 breaths per minute and the group with the body weight exceeding 2500 g: 54 breaths per minute. The differences between individual groups were statistically significant [31]. Studies on the importance of neonatal stabilization of critically ill newborns during transport carried out by Xu et al. (2019) showed that the study group of high risk of transported neonates was dominated by male patients whose average gestational age was 32.63 weeks and body weight was 1828.7 g. The control group of high-risk transported newborns was equal in terms of gender distribution, and the average gestational age was higher, 34.89 weeks, as well as body weight, 2397 g [32]. McAdams et al. (2008) presented studies on long-distance airborne transport of infants with very low birth weight and pneumoperitoneum. They showed that male newborns were dominant both in the group with pneumoperitoneum and the one with the babies without pneumoperitoneum. The birth weight of the patients with pneumoperitoneum was higher compared to the group of patients who did not suffer from it, whereas at the time of transport the infants without pneumoperitoneum were older and had higher body weight compared to the infants with pneumoperitoneum [33]. According to the authors’ own study, the preterm infants were older at the time of transport, whereas the group of full-term newborns was characterized by a higher body weight during transport and a more severe clinical condition on the basis of NACA scale. According to the source literature, airborne transport of newborns by highly specialized teams did not cause the deterioration their clinical condition. These conclusions were drawn from the results of vital signs measured before, during and after the transport, as well as from the standard evaluation carried out within 24 h after the transport [34,35,36,37].

In the further part of the study, the authors focused on the most frequently undertaken medical emergency procedures and medications used during neonatal transport. The analysis of the authors’ own study showed that the transported patients most often required intubation, mechanical ventilation and oxygen therapy. The most commonly used medications were antibiotics, prostaglandin E1, glucose, furosemide and vitamins. According to Yan Leung et al. (2019), nearly 70% of all transported patients required various procedures during transport. Furthermore, it was proved that the transported infants most often required assisted ventilation, oxygen therapy, inotropic support, sedation and prostaglandin infusion. Moreover, infants in whom complications occurred during transportation were more likely to require intubation and support with inotropic drugs [24]. In Abdulraheem et al.’s study (2016), analyzing cases of neonatal transports to tertiary care center at the University College Hospital (UCH) Ibadan in Nigeria, it was showed that only 3% of all newborns were transported in incubators, and during the transport oxygen therapy, administration of drugs such as adrenaline, dopamine and hydrocortisone as well as intravenous fluid therapy were most frequent [38]. However, the statistical analysis of the authors’ own study showed that in the group of premature infants, glucose and vitamins were used significantly more often, whereas in the group of full-term newborns the use of prostaglandin E1 and furosemide was more frequent.

According to the assumptions regarding the treatment of newborn children requiring specialist care that cannot be obtained in the center where they were born, it is necessary to transport them to the center that will provide comprehensive and multi-faceted diagnostic and therapeutic procedures as well as appropriate care [10,17,18,19,20]. This assumption is supported by the results of the authors’ own study, the case in which the patients in incubators were transported to the center with a higher level in the referral system, most often a tertiary one, and these were mainly medical transports. In turn, further analysis of the results of the authors’ own study showed that in the case of full-term newborns, rescue airborne transport was more frequent, and a patient was accompanied by a parent during the transport. The source literature emphasizes that family-oriented care is a mutual partnership between care providers and patients and their families. This model is not an option but should be a standard in neonatal/pediatric patient care. In the case of specialized neonatal transport in the context of patient-centered care, the presence of the family during such transport is taken into account [39,40]. Joyce et al.’s study (2015) on the family-focused care in the case of transporting ill pediatric patients showed that parents were more likely to accompany their children in the case of ground transportation than the airborne one [40].

The study on the contribution of the Polish Medical Air Rescue in the rescue procedure in case of severe upper limb injuries was conducted by Gałązkowski et al. (2014). Their study showed that the average inter-hospital air transport time of patients with severe upper limb injuries was 75.8 min and the total transport time—172.3 min [41]. Wejnarski et al. (2019) characterized airborne transport, both between hospitals and directly from the site of the event of patients with acute myocardial infarction or acute trauma in Poland between 2011 and 2016. They found that for patients with acute coronary syndrome, the median time that passed from the receipt of the call to patient transfer to the target hospital was 81 min, and the transport distance was 59.4 km, whereas in patients with sudden injury the median time from the receipt of the call to patient transfer to the target hospital was 115 min, and the distance was 135.9 km [42]. The results of the authors’ own study showed that the average time from the receipt of the call to patient transfer in the incubator to the target center was much longer and amounted to 269.16 min, while the distance during transport was also greater, that is 304.27 km, compared to the above analyses of air transports [41,42].

This study is the first retrospective analysis of patient transports in incubators performed by Helicopter Emergency Medical Service and Emergency Medical Service of the Polish Medical Air Rescue between 2012 and 2018. We would like to point out that the study has some limitations. The analysis covers only the information contained in the documentation of HEMS and EMS crews, without any knowledge of the further management or survival of the examined patients. However, these limitations do not affect the quality of the study. Further research on neonatal transport is necessary in order to comprehend this problem better and to provide the best and highest quality of care for the youngest patients who require transportation.

## 5. Conclusions

Among the patients transported in incubators by the HEMS and EMS air ambulance crews between 2011 and 2018 there were mainly males who, on the basis of the date of delivery, were qualified as full-term newborns. Most of the neonatal transports were carried out by an airplane as they were medical transports to the tertiary care center in the referral system.

Significant differences between preterm infants and full-term newborns were found in the case of age and body weight at the time of transport, diagnosis based on the ICD classification 10, the most commonly used medications (prostaglandin E1, glucose, furosemide, vitamins), NACA scale value as well as the type of mission and presence of an accompanying person during the transport.

Among preterm infants, the odds ratio is higher if there is a possibility of congenital malformations of the circulatory, gastrointestinal, nervous and respiratory system originating in the perinatal period.

## Figures and Tables

**Figure 1 ijerph-17-00705-f001:**
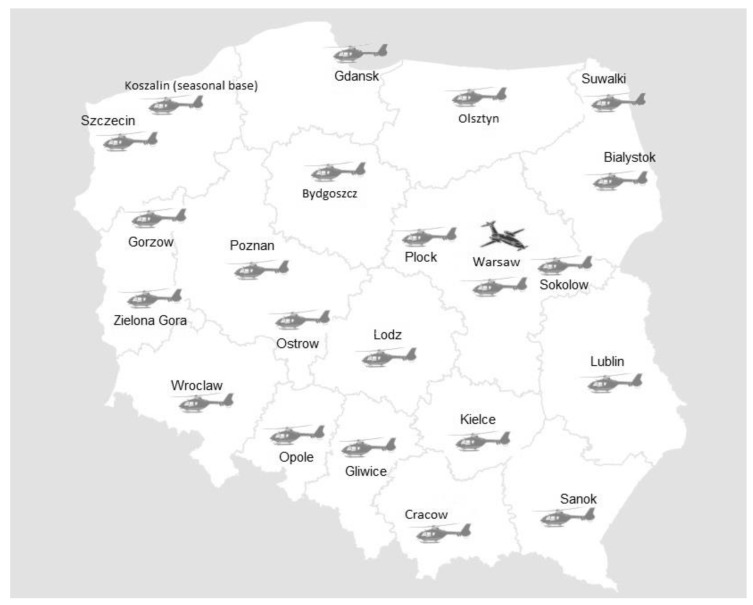
Location of Emergency Medical Service (EMS) and Helicopter Emergency Medical Service (HEMS) bases in Poland.

**Table 1 ijerph-17-00705-t001:** Characteristics of patients transported in incubators and statistical analysis of classification of patients based on the date of delivery vs. patient sex, clinical condition and most frequent medical emergency procedures and medications.

Variable	Total	Classification of Patient Based on the Date of Delivery		*p*-Value
		Preterm Infant	Full-Term Newborn	
Sex—*n* (%)				
Female	196 (44.95)	93 (47.45)	103 (52.55)	0.4403
Male	240 (55.05)	105 (43.75)	135 (56.25)	
Patient classification based on the date of delivery—*n* (%)				
Preterm infant	198 (45.41)	-	-	-
Full-term newborn	238 (54.59)	-	-	
Age during transport (days)—M (SD)	37.53 (43.53)	50.40 (44.49)	26.82 (39.73)	0.0000
Body weight during transport (g)—M (SD)	3121.18 (802.64)	2609.07 (702.68)	3547.24 (608.78)	0.0000
Diagnosis based on ICD classification 10—*n* (%)				
Congenital heart malformations	180 (41.28)	68 (34.34)	112 (47.06)	0.0270
Congenital malformations of the circulatory system	85 (19.50)	43 (21.72)	42 (17.65)	
Congenital malformations of the gastrointestinal system	60 (13.76)	34 (17.17)	26 (10.92)	
Respiratory disorders originating in the perinatal period	41 (9.40)	21 (10.61)	20 (8.40)	
Congenital malformations of the nervous system	18 (4.13)	12 (6.06)	6 (2.52)	
Congenital malformations affecting multiple systems	15 (3.44)	4 (2.02)	11 (4.62)	
Other diagnoses	37 (8.49)	16 (8.08)	21 (8.82)	
State of consciousness—*n* (%)				
Conscious	319 (73.17)	142 (71.82)	177 (74.37)	0.5337
Sedation	117 (26.83)	56 (28.28)	61 (25.63)	
Skin coloring—*n* (%)				
Normal	353 (80.96)	164 (82.83)	189 (79.41)	0.3415
Pale	40 (9.17)	19 (9.60)	21 (8.82)	
Yellowish	43 (9.86)	15 (7.58)	28 (11.76)	
Cyanosis—*n* (%)				
Yes	44 (10.09)	23 (11.62)	21 (8.82)	0.3351
No	392 (89.91)	175 (88.38)	217 (91.18)	
Respiratory effort—*n* (%)				
Present	52 (11.93)	23 (11.62)	29 (12.18)	0.6889
No	341 (78.21)	158 (79.80)	183 (76.89)	
Apnea	43 (9.86)	17 (8.59)	26 (10.92)	
Breath count (breaths per minute)—M (SD)	36.36 (16.95)	36.26 (16.03)	36.44 (17.71)	0.7738
Saturation (%)—M (SD)	93.55 (6.49)	93.99 (6.36)	93.18 (6.59)	0.1296
Heart rate (beats per minute)—M (SD)	141.86 (17.27)	141.18 (16.94)	142.44 (17.55)	0.9016
Systolic pressure (mmHg)—M (SD)	76.60 (14.56)	75.36 (13.33)	77.63 (15.46)	0.2056
Diastolic pressure (mmHg)—M (SD)	45.23 (10.62)	44.46 (9.58)	45.87 (11.39)	0.3549
NACA—M (SD)	3.66 (0.80)	3.57 (0.80)	3.74 (0.79)	0.0228
* Medical Emergency Procedures—*n* (%)				
Oxygen therapy	65 (14.91)	35 (17.68)	30 (12.61)	0.1388
Mechanical ventilation—respirator	75 (17.20)	34 (17.17)	41 (17.23)	0.9879
Intubation	119 (27.29)	55 (27.78)	64 (26.89)	0.8360
** Medications during transport—*n* (%)				
Antibiotics	131 (30.05)	55 (27.78)	76 (31.93)	0.3461
Prostaglandin E1	144 (33.03)	41 (20.71)	103 (43.28)	0.0000
Glucose	192 (44.04)	99 (50 00)	93 (39.08)	0.0222
Furosemide	63 (14.45)	20 (10.10)	43 (18.07)	0.0185
Vitamins	78 (17.89)	50 (25.25)	28 (11.76)	0.0003
Parenteral nutrition	60 (13.76)	31 (15.66)	29 (12.18)	0.2948

Notes: *—most frequent medical emergency procedures during patient transport (the results do not add up); **—most frequent medications during patient transport (the results do not add up); M — mean; SD — standard deviation; ICD-10 — the International Statistical Classification of Diseases and Related Health Problems; NACA — the National Advisory Committee for Aeronautics scale.

**Table 2 ijerph-17-00705-t002:** Mission characteristics.

Type of Aircraft—*n* (%)	
Helicopter	67 (15.37)
Airplane	369 (84.63)
Type of mission—*n* (%)	
Rescue transport	90 (20.64)
Medical transport	346 (79.36)
Season of the year—*n* (%)	
Spring	126 (28.90)
Summer	110 (25.23)
Autumn	103 (23.62)
Winter	97 (22.25)
Days of transport—*n* (%)	
Weekdays	381 (87.39)
Weekend	55 (12.61)
Referral level of target hospital—*n* (%)	
II level	95 (21.79)
III level	341 (78.21)
Accompanying person during transport—*n* (%)	
Parent	186 (42.66)
No accompanying person	250 (57.34)
Time of transport (min)—M (SD)	49.92 (27.70)
Time from taking care of patient to patient transfer to target hospital (min)—M (SD)	86.29 (33.57)
Time from receipt of call to patient transfer (min)—M (SD)	269.16 (118.55)
Distance (km)—M (SD)	304.27 (93.05)

M — mean; SD — standard deviation.

**Table 3 ijerph-17-00705-t003:** Classification of patients based on the date of delivery vs. mission characteristics.

Variable	Classification of Patients Based on due Date		*p*-Value
	Preterm Infant	Full-Term Newborn	
Type of mission—*n* (%)			
Rescue transport	29 (14.65)	61 (25.63)	0.0048
Medical transport	169 (85.35)	177 (74.37)	
Referral level of target hospital—*n* (%)			
II level	40 (20.20)	55 (23.11)	0.4641
III level	158 (79.80)	183 (76.89)	
Accompanying person during transport—*n* (%)			
Parent	72 (36.36)	114 (47.90)	0.0153
No accompanying person	126 (63.64)	121 (50.84)	
Time of transport—M (SD)	51.07 (27.87)	48.97 (27.59)	0.1950
Time of taking care of patient—M (SD)	86.10 (31.69)	86.45 (35.12)	0.2660
Time from receipt of call to patient transfer—M (SD)	258.58 (108.74)	278.07 (125.77)	0.1231
Distance—M (SD)	306.40 (94.18)	302.51 (92.28)	0.8150

M — mean; SD — standard deviation.

**Table 4 ijerph-17-00705-t004:** The occurrence of congenital malformations in surveyed patients depending on patient classification based on the date of delivery and sex.

Diagnosis—*n* (%)	Classification of Patient Based on the Date of Delivery				Sex			
	Preterm Infant	Full-Term Newborn	Odds Ratio (OR)	95% CI	Female	Male	OR	95% CI
Congenital heart malformations	68 (34.34)	112 (47.06)	0.59	0.40–0.87	80 (40.82)	100 (41.67)	0.97	0.66–1.42
Congenital malformations of the circulatory system	43 (21.72)	42 (17.65)	1.30	0.81–2.08	35 (17.86)	50 (20.83)	0.83	0.51–1.34
Congenital malformations of the gastrointestinal system	34 (17.17)	26 (10.92)	1.70	0.98–2.93	29 (14.80)	31 (12.92)	1.17	0.68–2.02
Respiratory disorders originating in the perinatal period	21 (10.61)	20 (8.40)	1.29	0.68–2.46	18 (9.18)	23 (9.58)	0.95	0.50–1.82
Congenital malformations of the nervous system	12 (6.06)	6 (2.52)	2.50	0.92–6.77	8 (4.08)	10 (4.17)	0.98	0.38–2.53
Congenital malformations affecting multiple systems	4 (2.02)	11 (4.62)	0.43	0.13–1.36	7 (3.57)	8 (4.17)	1.07	0.38–3.02
Other diagnoses	16 (8.08)	21 (8.82)	0.91	0.46–1.79	19 (9.69)	18 (7.50)	1.32	0.67–2.60

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
