# Peer review of "Neonatal Transport in the Practice of the Crews of the Polish Medical Air Rescue: A Retrospective Analysis"

_ijerph, 2020, doi:10.3390/ijerph17030705_

Round 1
Reviewer 1 Report
A very good work describing the Neonatal air transport.There is only a point I would like to point out :
In order for the reader to understand more the frame of HEMS system function in Poland, some information could be added about the space distribution of air bases of Helicopters and airplanes and the usual destination hospitals.( E.g. a map displaying those information).
Author Response
Dear Reviewer,
Thank you very much for the positive review of our paper.
In the article’s introduction we added the information on the activity of HEMS in Poland as well the map with the locations of HEMS bases in Poland – “Today, Polish Medical Air Rescue consists of 21 regional HEMS bases, 1 seasonal HEMS base (in Koszalin) and 1 EMS transport team located at the Warsaw Frederic Chopin Airport (Figure 1)”.
We didn’t take into consideration the locations of the hospitals, which the patients in incubators were most frequently transported to, as this was not included in the study design or in the material analysis and therefore the map would be confusing.
Reviewer 2 Report
Tabelle 1 und 3 zusammenfügen
Dear authors,
Congratulations for a very good publication about neonatal transport in Poland. The issue is up-to-date, because transporting by HEMS, or EMS of vulnerable full-term newborns and premature babies with small weight and in poor condition can be very challenging for a medical personal.
The authors examined 436 cases throughout the country in the period of 7 years.
I have a few comments and possible ideas on how to improve this work, or future research in this field:
1. Can you explain how are transported in Poland babies under weight analysed in the publication?
2. Do you have information about follow-up, health status of babies in destination hospital?
3. In my point of view an information’s in table no. 1 and 3 the authors can put together.
Author Response
Dear Reviewer,
Thank you very much for the positive feedback on our paper. We will use your suggestions in the future.
I have a few comments and possible ideas on how to improve this work, or future research in this field:
Can you explain how are transported in Poland babies under weight analysed in the publication?We added the information on the transport of neonates in the introduction according to the Reviewer’s suggestions – “Newborn transport is carried out in a direct mode, namely bed-to-bed (B2B) mode, in which the patient is picked up directly from the ordering hospital and handed over in the target hospital in the same way, and in combined mode (nB2B), in which HEMS crews cooperate with ground-based teams.”
Do you have information about follow-up, health status of babies in destination hospital?
The aim of our study was to concentrate on presenting the characteristics of patients transported in incubators by HEMS crews in Poland, as well as on the missions themselves, and the information concerning the future of these patients wasn’t the aim of this paper.
In my point of view an information’s in table no. 1 and 3 the authors can put together.
According to the Reviewer’s suggestions Table 1 and Table 3 have been combined and the numeration of the remaining tables has been changed.